# Optimization of Cohesive Parameters in the Interfacial Transition Zone of Rubberized Concrete Based on the Response Surface Method

**DOI:** 10.3390/polym16111579

**Published:** 2024-06-03

**Authors:** Kai Min, Xianfeng Pei, Houmin Li, Zhou Cao, Zijiang Yang, Dingyi Hao, Wenchao Li, Cai Liu, Keyang Wu

**Affiliations:** 1School of Engineering, Architecture and the Environment, Hubei University of Technology, Wuhan 430068, China; 102200768@hbut.edu.cn (K.M.); 102100805@hbut.edu.cn (X.P.); haodingyi1015@163.com (D.H.); a424185749@163.com (W.L.); c1357962470@163.com (C.L.); 2China Construction Third Bureau First Engineering Co., Ltd., Wuhan 430040, China; csceczj319025@163.com (Z.C.); zjsjygszh2023@163.com (Z.Y.); 3Wuhan Construction Engineering Co., Ltd., Wuhan 430056, China; wukeyang@wceg.com.cn

**Keywords:** rubber concrete, ITZ, cohesive parameters, RSM, numerical simulation

## Abstract

Rubber concrete has been applied to a certain extent in fatigue-resistant structures due to its good durability. Based on a cohesive model of rubber composed of a five-phase material containing mortar, aggregate, rubber, aggregate-mortar interfacial transition zone (ITZ), and rubber-mortar ITZ, this paper studies the influence of the cohesive parameters in the rubber-mortar ITZ on the fatigue problem of rubber concrete on the mesoscopic scale. As the weak part of cement-based composite materials, the ITZ has a great influence on the mechanical properties and durability of concrete, but the performance of the ITZ is difficult to test in macro experiments, resulting in difficulties in determining its simulation parameters. Based on the cohesive model with a rubber content of 5%, this study uses Monofactor analysis and the Plackett-Burman test to quickly and effectively determine the primary and secondary influences of the cohesive model parameters in the rubber-mortar ITZ; further, the response surface method is used to optimize the cohesive parameters in the rubber-mortar ITZ, and the numerical simulation results after optimizing the cohesive parameters are compared and analyzed with the simulation results before optimization. The results show that, under the setting of the optimized parameters, the simulation results of each item of the optimal cohesive model parameters in the rubber-mortar ITZ are in line with the reality and closer to the experimental data, and they are also applicable to rubber concrete models with different rubber dosing.

## 1. Introduction

Rubber concrete is prepared by replacing the aggregate in ordinary concrete with rubber crumb. Compared with ordinary concrete, the incorporation of rubber reduces the mechanical properties [1,2,3,4] but improves abrasion resistance, frost resistance, and durability [5,6,7,8]. The birth of rubber concrete has opened up a new direction, not only for the durability of concrete research but also for the recycling of waste tires to provide a novel and feasible way. Despite the many advantages of rubberized concrete, it is still in the research and development stage and has a long way to go before it can be applied on a large scale; it currently can only be used on a small scale. Currently, the research on rubber concrete mainly focuses on the preparation process, performance characterization, fine microstructure, and material durability. The current stage of rubber concrete research is mainly focused on experimental aspects, and its mechanical properties, structural composition, and damage form are obtained by experiments and experience, subject to the experimental environment, experimental personnel, and other external factors, the results of the dispersion is relatively large. However, with the development of science and technology and the improvements in computer technology, the research means of concrete are no longer limited to experimental research, and the method of numerical simulation has gradually become the main research method [9]. In recent years, the theory of numerical simulation has also been in rapid development and progress, involving the discrete element model [10], finite element model [11], lattice model [12], and near-field dynamics model [13,14]. In particular, finite element models have developed extraordinarily rapidly, and the span of research scales is so large that accurate simulation renditions can be performed from mesoscale concrete models [15] to macroscopic concrete models [16]. Scholars [17,18] have analyzed rubber concrete from the micro-scale ($10^{-8}–10^{-4}\;m$), mesoscale ($10^{-4}–10^{-1}\;m$), and macro-scale ($>10^{-1}\;m)$; for a conventional numerical study, the micro-scale is too small to study the effect of its molecular composition on the overall mechanical properties with the current stage of the means, while the macro-scale study of rubber concrete will ignore the effect of rubber on the concrete, which loses the significance of the study. Additionally, the macro-structure in the stage of numerical simulation will produce macroscopic strain and macroscopic stress concentration in the numerical simulation stage. Since then, it has been of great significance to study the mesoscale of rubberized concrete, which can take into account both the influence of rubber properties on concrete and the interaction between aggregates and rubber particles.

Over the past two decades, concrete mesoscale modeling studies have successfully established aggregate, mortar, admixture, and ITZs [19] where each constituent interacts with each other through mechanical relationships, thus affecting the overall structural strength. On the fine scale, the study of ITZs appears to be very difficult, and their actual thicknesses are usually in the range of 10–50 um, which is beyond the scope of mesoscale studies [20]. Victor [21] suggested that all ITZs in concrete can be represented by elements of zero thickness. In recent years, with the development of research techniques, the method of cohesive modeling [22] has been proposed to simulate the damage of very small thickness units. Dugdale [23] first proposed the cohesive force model for the study of crack expansion in large thin plates. Subsequently, a large number of researchers applied the cohesive model to the ITZ of concrete. Wang [24] developed a tensile and compressive model for mortar to calibrate the parameters of the cohesive model, and it was found that the performance of the ITZ was affected by the normal and tangential strengths of cohesive elements as well as the fracture energy. With the increasing use of cohesive modeling, the original cohesive damage theory and law can no longer meet the needs of different situations, and thus different damage forms have been developed. From the basic bilinear model to several hybrid cohesive models [25,26], Dong et al. [27] proposed an improved cohesive model for the concrete fracture to capture the complex fracture process inside concrete. The improved principle is based on near-field dynamics, where the strength and fracture energy of the transition zone between the mortar and the interface are characterized by the nature of each nearby failed bond, making it easy to determine the relevant parameters in the model. Turon [28] derived a new cohesive constitutive equation to define the correspondence between interlayer strengths and demonstrated that the load–displacement curves obtained using cohesive elements were identical to the load solutions in the hybrid model. Dimitri et al. [29] parametrically analyzed the effect of different loading paths on the displacements in the cohesive zone, the energy loss, and the failure region of the hybrid mode and found that the error of the conventional hybrid mode was found in some specific cases. The original cohesive model was modified by introducing normal, tangential, and coupled damage variables to improve the accuracy of the hybrid model.

The concrete damaged plasticity (CDP) model was first proposed by Lubliner [30]. The characteristic of this model is that it defines the states of damage and stiffness recovery when the concrete is plastically damaged. When it is applied under cyclic loads, it can well deduce the mechanical behavior of component damage and failure. Based on the CDP model, the research group adopted a numerical model of three-point bending rubberized concrete cohesive without precast cracks with random aggregates to study the fatigue simulation of rubberized concrete members.

The goal of this study is to analyze and optimize the cohesiveness parameters of the rubber-mortar interface based on the above numerical model of cohesiveness. By using Monofactor analysis, the Plackett-Burman test, and the response surface method, a set of optimized rubber-mortar interface cohesive parameters are obtained. After being verified in many aspects, the optimized simulation results of the cohesiveness parameters of the rubber-mortar interface are in line with the practice and are closer to the experimental data, which can provide a reference for the subsequent research of the mesoscale model, including the rubber-mortar interface, and is of great research significance.

## 2. Model Building

### 2.1. Detailed Model Geometry Generation

In the mesoscale study, rubber concrete can be regarded as a five-phase composite material composed of mortar, aggregate, rubber, aggregate-mortar ITZ, and rubber-mortar ITZ. Previous studies have pointed out the effects of aggregate size, particle size distribution, aggregate shape, and interfacial zone on the properties of concrete. Zhong [31] investigated the effect of aggregate shape on the numerical analysis results of the fine view model. The results showed that the circular aggregate model is optimal for numerical simulation. Therefore, the circular aggregate model is followed in this study.

Concrete performance is optimized when aggregates are distributed according to Fuller’s gradation [32]. However, this study is at the two-dimensional level, and it is necessary to process the three-dimensional Fuller grading formula with the Walraven formula [33] to obtain the two-dimensional optimal particle size distribution curve. The formula is as follows:(1)$$P_{c}\left(D<D_{0}\right)=P_{k}(1.065D_{0}^{0.5}D_{max}^{-0.5}-0.053D_{0}^{4}D_{max}^{4}-0.012D_{0}^{6}D_{max}^{-6}-0.0045D_{0}^{8}D_{max}^{-8}-0.0025D_{0}^{10}D_{max}^{-10})$$
where $P_{c}$ represents the percentage of aggregate area with size D less than $D_{0}$, $P_{k}$ represents the percentage of aggregate area in the total area. In this study, $P_{k}$ is 0.7, $D_{max}$ represents the diameter of the maximum aggregate size, and the maximum diameter is 20 mm. In this study, the cohesive model of rubberized concrete with 5% rubber admixture was used, and the model size was taken as 150 mm×550 mm. The corresponding aggregate size distribution was calculated according to Equation (1), as shown in the following Table 1.

Based on the above theories and data, the random generation of corresponding size aggregates is achieved in Python, and then the corresponding property values are assigned to each component to achieve the effect of simulating the real rubberized concrete aggregates in a fine view. The rubber concrete beam with 5% rubber admixture is shown in Figure 1, where gray represents mortar, red represents aggregate, and black represents rubber.

### 2.2. Constitutive Model of Concrete

In this study, the constitutive model of concrete adopts the CDP model. The mortar can be regarded as a lower-strength type of concrete, and its constitutive law uses the CDP model. Aggregates and rubbers are considered homogeneous elastomers. Under cyclic loading, the mechanical performance degradation mechanism of the model is extremely complex and involves the difference between the damaged units and the undamaged units. When the load changes from tension to compression, the tension-damaged units will not completely lose the mechanical bearing capacity. Instead, due to the compressive load, the cracks will close, resulting in the situation of recovery of compressive stiffness. The model assumes that the elastic modulus E is represented by the scalar degradation variable d, and the formula is:(2)$$E=\left(1-d\right)E_{0}$$
where $E_{0}$ presents the initial elastic modulus of the material. The formula is valid on both the tensile side $\left(\sigma_{11}>0\right)$ and the compressive side $\left(\sigma_{11}<0\right)$ under cyclic loading. The stiffness degradation variable has damage variables of $d_{t}$ and $d_{c}$ in uniaxial tension and compression, and the damage manifestation form under cyclic loading is assumed to be a formula in the software as:(3)$$s_{t}=w_{t}r^{*}\left(\sigma_{11}\right),\;\;\;\;\;0\leq w_{t}\leq 1$$
(4)$$s_{c}=w_{c}r^{*}\left(\sigma_{11}\right),\;\;\;\;\;0\leq w_{c}\leq 1$$
(5)$$r^{*}\left(\sigma_{11}\right)=\left\{\begin{array}{c} 1,\;\;\;\;\;\;\sigma_{11}>0\\ 0,\;\;\;\;\;\;\;\sigma_{11}<0 \end{array}\right.$$
where $w_{t}$ and $w_{c}$ are the corresponding weighting factors that control the tensile stiffness and compressive stiffness when the loading direction is reversed.

### 2.3. Modeling of Cohesive Elements in ITZs

In this study, the ITZ is regarded as a zero-thickness unit, which retains the relevant mechanical properties of the actual ITZ to reach the accuracy of the simulation.

According to the random aggregate generated from the simulation in this study, to add a cohesive zone at the contact surface in aggregate and mortar, a layer of cohesive element with zero thickness is inserted, utilizing shared nodes to simulate the behavior of the ITZ. Based on Figure 2, the cohesive element created for the ITZ can be observed.

An ITZ is generally considered to be the weak part of cement-based composite materials [34], and its performance is similar to that of mortar. Researchers use the percentage of mortar performance to study and judge the performance of an ITZ. Xiao et al. [35] concluded that, with reduced compressive strength and an elastic modulus an ITZ of 20% than those of the mortar, Kim et al. [36] used 50% as the fracture energy of ITZ concerning the mortar, and, unlike Kim, Li et al. [37] concluded that the ITZ has a fracture energy equivalent to 80% of mortar. Although different researchers have different opinions in determining the mechanical properties of an ITZ, the ultimate goal is to try to determine the optimal values of ITZ-related parameters by a trial and error method. There are two types of ITZ in this study, the aggregate-mortar interface transition zone and the rubber-mortar transition zone. According to previous experience, the interfacial strength of a rubber-mortar ITZ is less than that of an aggregate-mortar ITZ, and the specific relevant parameters are listed in the following Table 2.

The data regarding the aggregate-mortar ITZ in Table 2 are derived from [24] and the data regarding the rubber-mortar ITZ use trial values.

### 2.4. Model Calculations

By comparing the simulation results of this model with the experimental results of peak load and fatigue load in the literature [38], the feasibility and accuracy of this model have been verified in the previous study [39].

#### 2.4.1. Model Loads

The model is solved using the ABAQUS (2021 Version, Dassault systemes, Paris, France). A three-point bending test places the specimen on two support points at a certain distance and applies a downward load to the specimen at the midpoint of the two support points. When the three contact points of the sample form two equal torques, three-point bending will occur, and the sample will eventually break at the midpoint. In this study, the load loading point is in the middle of the upper part, and the supports are applied at the bottom 100 mm away from the boundary on both sides, as shown in Figure 3.

Firstly, the peak load of the model is calculated by applying a displacement support of 2 mm at the upper load loading point and stopping the calculation when the model calculation does not converge to obtain the peak load $F_{max}$ of the model. Subsequently, the fatigue life of the model is calculated for different stress levels, still using the same model, with cyclic centralized force constraints applied at the upper load loading point. The loads are applied in sizes ranging from the minimum load $P_{min}$ to the maximum load $P_{max}$, where $P_{min}/P_{max}$ = 0.1 and the fatigue load stress level S = $P_{max}/F_{max}$. When studying fatigue performance, a series of representative stress levels, such as 0.95, 0.85, 0.75, and 0.65, can be selected. In this study, the stress levels S were taken as 0.85 and 0.75 for the study. These two stress levels are also consistent with the design of the comparative experiment.

#### 2.4.2. Peak Load

The experimental and simulation results of peak loads under three-point bending loads for this model are summarized in Table 3. Moreover, comparing the peak loads of the tests and simulations at the same stress level, the simulation results are in good agreement with the test results, with a maximum absolute error of 1.88%.

#### 2.4.3. Fatigue Life

The experimental and simulation results of rubber concrete at stress levels S = 0.85 and S = 0.75 are summarized in Table 4. Since the fatigue life results are relatively discrete, only the average values of minimum life and maximum life are taken as references in the experimental results.

#### 2.4.4. Forms of Damage

In this simulation, the damage form of rubberized concrete is shown through the form of stiffness damage. There are two forms of damage: one in the form of static pressure damage and the other in the form of fatigue damage. The scalar stiffness degradation (SDEG) contour map visualizes the damage. generated by Abaqus. Since the damage occurs in the middle of the span, to facilitate the observation, the position in the middle of the beam is taken as shown in the black box in Figure 4, and its size is 150 mm×150 mm. The subsequent screenshots are obtained according to this method.

## 3. Cohesive Parameter Optimization

### 3.1. Monofactor Analysis

Monofactor analysis can determine the relevant parameter values quickly with fewer tests on a small range of parameter choices. In this study, a rubberized concrete model with 5% rubber admixture is taken as the object of study, and the differences are reflected by comparing the two numerical results of the peak load at the time of model damage and the fatigue load at 0.85 stress level.

The six factors of cohesive parameters analyzed in the Monofactor analysis include normal modulus, tangential modulus, normal strength, tangential strength, normal fracture energy, and tangential fracture energy. Among them, each factor is assigned three values for calculation, and another set of standardized groups is established; the corresponding information is shown in the following Table 5 and Table 6.

Each parameter of the factors in the standard group was changed in turn, and the values of peak load at damage and fatigue load at 0.85 stress level were compared and observed. The results are shown in Figure 5.

As shown above, a single change in the value of a factor does not result in a monotonically increasing or decreasing value of the peak load at damage and fatigue load at the 0.85 stress level of the model. This is because, in the fatigue simulation of rubberized concrete, the rubber-mortar interface does not rely on a single mechanical parameter but rather on a series of mechanical parameters acting together on the overall member. Therefore, it is necessary to investigate the interaction mechanism between them.

The literature experiments show that the peak load of rubber concrete with 5% rubber admixture is 25.05 kN, its minimum fatigue life value is 1485, and its maximum fatigue life value is 5883 load cycles at 0.85 stress level. In order to ensure the accuracy of the optimization of the cohesive parameters at the rubber-mortar interface, according to the fitting degree between the numerical model and the experiment, at the same time, it is also necessary to consider the corresponding result contour map of the numerical model to finally determine the optimized value of the cohesive parameters at the rubber-mortar interface. The following Figure 6 and Figure 7 list the damage forms of static pressure and fatigue loads when the normal modulus is 25,000 MPa and 35,000 MPa, respectively.

As can be seen from Figure 6 and Figure 7, a single change in the normal modulus parameter of the modeled rubber-mortar interface results in a change in the peak load and fatigue life of the overall member as well as its damage form. However, in Figure 6, two distinct damage bands appear in the resultant contour map at a normal modulus of 35,000 MPa, which is clearly different from the result of one penetrating damage band in most of the conventional three-point bending static pressure loading experiments. Therefore, such optimization parameters that lead to abnormal results can be discarded by combining the resultant contour map of the model, and the accuracy of the optimization parameters can be further improved.

In summary, after considering the peak load magnitude, the load cycles of fatigue life, and the damage forms, the range of values for the selection of rubber-mortar interface cohesive parameters is shown in Table 7.

### 3.2. Plackett–Burman Design

The Plackett–Burman design is a method for pre-test analysis which is suitable for quickly and efficiently selecting the most important factors from a large number of factors under examination. It can estimate the main effect of factors as accurately as possible with the minimum number of trials, and only two levels (minimum and maximum, as shown in Table 7) are set for each factor. The order of parameter selection for the test runs forms a specific orthogonal table or a list of near-orthogonal tables, ensuring balance and comparability among the individual variable factors.

Treating the six cohesive parameters studied above as independent variables and the peak load of model damage and the fatigue life at 0.85 stress level as two response variables, 12 sets of simulation tests were designed by using the Plackett–Burman module in the Design-Expert, and the results are shown in Table 8.

As can be seen from Table 8, among the 12 groups of rubber-mortar interface cohesive parameters, the peak load varied from 24.96 kN to 25.4 kN, with a smaller variation and a difference of 1.76%, while the fatigue life varied from 1744 to 9733 load cycles, with a larger variation and a difference of 458.1%. It can be seen that the rubber-mortar interface cohesive parameters have a significant effect on the fatigue life and the effect on the peak load is less pronounced, which is consistent with the law derived from the experimental results, this also precisely explains that the calculation results of ordinary mechanical simulation (tensile, compressive and flexural) calculations are not sensitive to the changes in the rubber-mortar interface cohesive parameters. In addition, a few studies have shown that the modeled fatigue life results exhibit a certain degree of dispersion.

Based on the data in Table 8, an analysis of variance (ANOVA) was performed to calculate the statistical values of the relevant variables, such as a sum of squares, independent degrees of freedom, and mean squares, from which the coefficients of influence of each of the rubber-mortar interfacial cohesive parameters on the peak load, as well as the fatigue life of the overall member, were determined. The ANOVA statistics are shown in Table 9 and Table 10.

In the Packett–Burman design, the coefficient of influence of a parameter of the independent variable is determined by the *p*-value in the analysis of variance (ANOVA). If *p* < 0.0001, the effect of the parameter is considered extremely significant; if *p* > 0.05, the effect of the parameter is considered insignificant. As can be seen from the table, in the peak load ANOVA, the normal modulus and tangential modulus have extremely significant effects on it, the normal strength, tangential strength, and normal fracture energy have significant effects on it, and the tangential fracture energy has no significant effect on it. In the fatigue life ANOVA, the normal modulus and tangential modulus have extremely significant effects on it, the tangential strength has significant effects on it, and the normal strength, normal fracture energy, tangential fracture energy has no significant effects on it. Additionally, normal strength, normal fracture energy, and tangential fracture energy have no significant effect on it. In the comprehensive analysis, normal modulus and tangential modulus have relatively large effects on the peak load of model damage and fatigue life at 0.85 stress level, while other parameters have relatively small effects.

To correctly reflect the relationship between the model parameter variables and the dependent variables (peak load and fatigue life), and to ensure that the statistical data are meaningful, the residual normal distribution of the dependent variable is plotted according to the data in the above table, as shown in Figure 8.

Where the vertical coordinate is the normal probability and the horizontal coordinate is the internal studentized residuals. The results show that the dependent variable error of ANOVA is small, confirming its validity.

In summary, the Packett–Burman test analysis determined that the main influencing factors in the rubber-mortar interface parameters were normal modulus, tangential modulus, normal strength, and tangential strength, and the secondary influencing factors were normal fracture energy and tangential fracture energy.

### 3.3. Response Surface Method Optimization

The aforementioned study confirmed the main influencing factors in the rubber-mortar interfacial cohesive parameters, which were further refined and optimized. The normal fracture energy and tangential fracture energy were taken as values of 0.028 N/mm and 0.084 N/mm, respectively, according to the original model parameters. For each of the four main influencing factors, three-level factors were selected, and the test parameters were designed as shown in Table 11.

According to the four test cohesive parameters designed in Table 11 as the independent variables, the peak load of model damage and the fatigue life of 0.85 stress level are still taken as the dependent variables, and the Box–Behnken module in the design software Design-Expert (10.0.1 Version) is utilized to carry out the test. The results are as follows in Table 12.

In Table 12, it can be seen that the variation range of peak load is from 24.974 kN to 25.317 kN, with a small variation and a difference of 1.37%; the variation range of fatigue life is from 2499 to 6860 load cycles, with a large variation and a difference of 174.51%.

Based on the data in Table 12, ANOVA was carried out to calculate the statistical values of the relevant variables, such as the sum of squares, independent degrees of freedom, and mean square, and thus the coefficients of influence of the four main influencing factors on the peak load as well as the fatigue life of the overall member. The statistical table of ANOVA and the calculation of quadratic correlation terms for all the parameters are shown in Table 13 and Table 14.

Among them, F indicates the influence coefficient of the dependent variable where the relevant parameter is located, and the larger the value is, the more significant the influence is. The *p* value determines the influence coefficient of the independent variable parameter, and if *p* < 0.0001, it is considered that the influence of the parameter is extremely significant. If 0.0001 < *p* < 0.05, it is considered that the influence of the parameter is more significant, and if *p* > 0.05, it is considered that the influence of the parameter is not significant. In the peak load ANOVA, normal modulus and normal strength had extremely significant effects, and tangential modulus and tangential strength had more significant effects. In the fatigue life ANOVA, the normal modulus and normal strength have a significant effect on it, and the tangential modulus and tangential strength have an insignificant effect on it. In the combined analysis, both normal modulus and normal strength have relatively large effects on the fatigue life at peak load and 0.85 stress level of model damage. In addition, the quadratic relationship between the independent variable parameters and the parameters is also considered in the table, compared to which there is no significant effect of the primary term single-factor parameters.

To correctly reflect the relationship between the model parameter variables and the dependent variables (peak load and fatigue life), and to ensure that the statistical data are meaningful, the residuals of the dependent variables are plotted with normal distribution based on the data in Table 13 and Table 14, as shown in Figure 9.

Similarly, the vertical coordinate is the normal probability and the horizontal coordinate is the internal studentized residuals. The results show good results that accurately reflect the significant role of normal modulus and normal strength influence among the four main influencing factors, while the tangential modulus and tangential strength influence is less significant.

Based on studies from the statistical perspective the influence of the cohesion parameters at the rubber-mortar interface on the peak load and fatigue life of the model failure, this research further considers the damage contour map of the model failure. Without considering the magnitude of the damage, only the rationality of its failure form is analyzed. The failure forms are divided into the failure form under static pressure load and the failure form under fatigue load. The damage contour map corresponding to 29 sets of design parameters in Table 12 is shown in Figure 10 and Figure 11.

As can be seen from Figure 10 and Figure 11, all the static pressure load damage forms and fatigue load damage forms are in line with the actual expectations. In the case of parameter changes, the static pressure load damage forms are basically the same, and the damage initiation position and the damage zone direction are basically the same; additionally, the fatigue load damage forms are not quite the same, and the damage initiation position and the damage zone direction are also changed. It can be seen that the change in the cohesive parameters of the rubber-mortar interface has a significant effect on the damage form of fatigue loading and has no significant effect on the damage form of static pressure loading.

### 3.4. Optimization of Rubber-Mortar Interface Parameters

Based on the results of the aforementioned experimental analysis, the optimization value of the rubber-mortar interface cohesive parameters is deeply refined to obtain simulation data that better meet the experimental expectations, and the optimization is carried out following the given objectives, taking into account the efficiency of the computer calculations as well as the changes in the rubber dosage. Based on the experimental data of the 5% rubber dosage, the optimization objective value of the peak load is 25.05 kN and the range of the fatigue life test is 1485–5883 load cycles. The average value of the optimized target was 3684 load cycles. The optimized rubber-mortar interface cohesive parameter design and the expected target value are shown in Table 15.

The simulation results of the optimized rubber-mortar interface cohesive parameters are shown in Table 16. The maximum error rate is 1.629%, which proves that the optimization of the rubber-mortar interface cohesive parameters is reasonable and effective, and this set of parameters can be applied to the study of rubber-concrete models.

## 4. Numerical Simulation of Rubber-Mortar Interface Cohesive Parameters after Optimization

The previous work defined the optimal rubber-mortar interface cohesive parameters as normal modulus of 29,300 MPa, tangential modulus of 11,000 MPa, normal strength of 2.57 MPa, and tangential strength of 8.6 MPa. To further explore the reliability and applicability of cohesive parameters, the rubber dosing of the original rubber concrete model was adjusted to 2.5%, 7.5%, and 10%, respectively, with other settings remaining unchanged. A comparison of the simulation results before and after optimization was made in terms of the peak load, fatigue life, and damage form of the model.

### 4.1. Peak Load after Optimization of rubber-mortar Interface Parameters

According to the simulation above, the peak loads of model damage were calculated for rubber dosing of 2.5%, 5%, 7.5%, and 10%, respectively, and the detailed comparison results are shown in Figure 12 and Table 17.

As can be seen from the above table, the simulation results after optimization of the rubber-mortar interface cohesive parameters are all closer to the experimental results, and the error is relatively small. The minimum error before optimization is 1.41% and the maximum error is 3.6%; the minimum error after optimization is 0.55% and the maximum error is 2.82%. The results demonstrate that the optimized rubber-mortar interfacial cohesive parameters can be applied to the peak load simulation of rubber-concrete models with different dosages and that the accuracy of the results is better.

### 4.2. Fatigue Life after Optimization of Rubber-Mortar Interface Cohesive Parameters

The fatigue life is calculated based on the peak loads of the different rubber dosing models mentioned above and then taking the stress levels of 0.85 and 0.75, respectively. Due to the discrete nature of the fatigue life in the experiment, the experimental error is relatively large, and there is no accurate life point, only a reasonable range of life. As a result, when performing simulation and experimental error analysis, the fatigue life of the experiment is taken as the average of the maximum and minimum values in the experiment as the experimental reference. The detailed comparison results are shown in the following Table 18 and Table 19, and Figure 13.

As can be seen from the above Table 18 and Table 19, when the stress level is 0.85, the minimum error after optimization of the rubber-mortar interface cohesive parameters is 2.33% and the maximum error is 9.97%; when the stress level is 0.75, the minimum error after optimization of the rubber-mortar cohesive interface parameters is 0.8% and the maximum error is 8.47%, which is better optimization and effectively improves the accuracy of the simulation results.

### 4.3. Damage Forms after Optimization of Rubber-Mortar Interface Cohesive Parameters

This study investigates the effect of rubber-mortar interface cohesive parameters on the peak load and fatigue life of model damage while considering whether the damage forms presented in the SDEG contour map in Abaqus are reasonable when the model is damaged (regardless of the size of the damage). The damage forms are categorized into static pressure load damage and fatigue load damage. The static pressure damage forms of the model with rubber dosing of 2.5%, 5%, 7.5%, and 10% and the fatigue damage forms with stress levels of 0.85 and 0.75 are shown in Figure 14, Figure 15 and Figure 16.

As shown in the above figures, after the optimization of the cohesive parameters of the rubber-mortar interface, the damage initiation positions of the static pressure and fatigue loads are basically the same in all models, there is only one damage zone, and the direction of the damage zone is also basically the same. It is proven that the damage forms of all optimized models match the actual requirements, and the effect of SDEG is also better.

## 5. Discussion

The present study is based on a zero-thickness cohesive damage model of the rubber-mortar interface, and the cohesive parameters of the rubber-mortar interface are optimized by Monofactor analysis, the Packett-Burman design, and the response surface method. The feasibility of the optimized rubber-mortar interface cohesive parameters is verified from the peak load, fatigue life, and damage form of the model, and the accuracy of the optimized rubber-mortar interface cohesive parameters is proved by comparing and analyzing the simulation results before and after the optimization of the rubber-mortar interface cohesive parameters.

### 5.1. Influence of Rubber-Mortar Interface Parameters

In this study, six kinds of rubber-mortar interface cohesive parameters were studied and analyzed. It was determined that the main influencing factors of rubber-mortar interface cohesive parameters were normal modulus, tangential modulus, normal strength, and tangential strength, while the secondary factors were normal fracture energy and tangential fracture energy.

The three-point bending component model studied in this study mainly focuses on the tensile failure in the middle of the component. The normal modulus reflects the ability of the material to resist normal deformation under the action of normal stress. When the material is subjected to tensile stress, the normal modulus has a significant effect on the peak load and fatigue load of the member. The tangential modulus reflects the ability of the material to resist the deformation caused by the relative slip of parallel sections under the action of shear stress. The model components in this study are mixed with a large amount of aggregate and rubber, and the strain properties of each component are different, while the relative displacement is easy to occur at the rubber-mortar interface. As a result, the tangential modulus parameter has a significant impact on the failure peak load of the model and the fatigue life at the stress level of 0.85. As for other parameters, they work together on the rubber-mortar interface and affect the crack width when the rubber-mortar interface is damaged. When studying the peak load of the model failure, the size of the crack broadband can represent the weak area of the rubber-mortar interface of the model, and the starting position of the model failure and the failure extension of the model will gather in the weak area, so these parameters also directly affect the peak load of the model failure. As the component model is mainly damaged by tension, the crack width is larger in the normal direction and smaller in the tangential direction, so the influence of the normal fracture energy and tangential fracture energy on the rubber-mortar interface is not obvious. When the fatigue life at the stress level of 0.85 is studied, the crack width is not representational, and the damage location can only be determined at the first time of damage. Since then, the crack has been in the open and closed cycle, and the final failure is affected by the modulus of the material. Therefore, the influence of these parameters on the fatigue life of 0.85 stress level is also insignificant.

### 5.2. Optimization of Rubber-Mortar Interface Cohesive Parameters

In this study, the target value of peak load optimization and fatigue life optimization of the model was set as 25.05 kN and 3684 load cycles in advance. The response surface method was used to analyze the rubber-mortar interfacial cohesive parameters of the rubber concrete model with 5% rubber admixture, focusing on optimizing the rubber-mortar interfacial cohesive parameters of the four main influences and finally arriving at a set of optimal data, as follows: the normal modulus is 29,300 MPa, the tangential modulus is 11,000 MPa, the normal strength is 2.57 MPa, and the tangential strength is 8.6 MPa. The cohesive model of rubberized concrete with different rubber dosages is simulated, and it is found that the error of the simulation results is smaller than the prediction results, which is closer to the real experimental data, and the accuracy of the numerical simulation is effectively improved.

### 5.3. Potential Applications and Developments

In this study, rubber-mortar interface cohesive parameters optimization is proven to be reasonable and effective, which provides a more accurate cohesive interfacial parameter reference for the subsequent rubber concrete ITZ performance research and improvement methods. It is also of reference significance for other polymer concretes as well as multi-facial concretes, especially for those like recycled concrete, which have both old and new ITZs. As the weak part of cementitious composites, the ITZ greatly affects the overall mechanical properties and durability of the material. The ITZ can better express the mechanical relationship between different substances to achieve a more accurate performance analysis.

This study is limited to the mechanical properties of the ITZ, and subsequent cohesive parameter optimization studies on the durability of concrete in terms of freeze–thaw cycles, ionic permeability, and salt-freeze multi-field coupling can be considered.

## 6. Conclusions

A new numerical model containing plastic damage theory and cohesive element in the ITZ is adopted.The response surface method is used to analyze and optimize the rubber-mortar interfacial cohesive parameters of the new numerical model.An optimal rubber-mortar interfacial cohesive parameter combination is obtained, featuring a normal modulus of 29,300 MPa, tangential modulus of 11,000 MPa, normal strength of 2.57 MPa, and tangential strength of 8.6 MPa.It is found that, after optimizing the cohesive parameters, error values of the peak load and fatigue life at different stress levels decrease significantly.The failure form conforms to the actual situation, and the effect of the SDEG contour map is also better.In the rubber concrete models with rubber contents of 2.5%, 7.5%, and 10%, errors between simulation and experiment results under the optimal setting of the cohesive parameters at the rubber-mortar interface are all within a reasonable range.This set of optimized cohesive parameters has fully verified its feasibility and wide applicability in the cohesive model of rubber concrete, which can effectively improve the accuracy of this type of numerical simulation results.

## Figures and Tables

**Figure 1 polymers-16-01579-f001:**
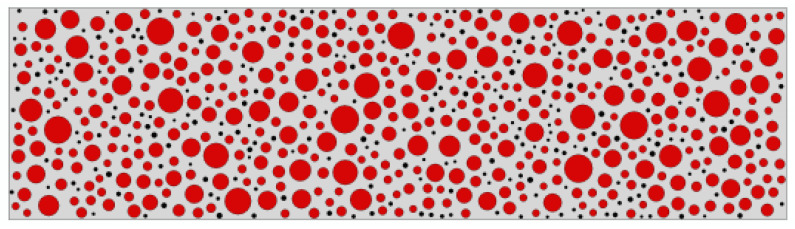
RC beam with 5% rubber admixture. Where gray represents mortar, red represents aggregate, and black represents rubber.

**Figure 2 polymers-16-01579-f002:**
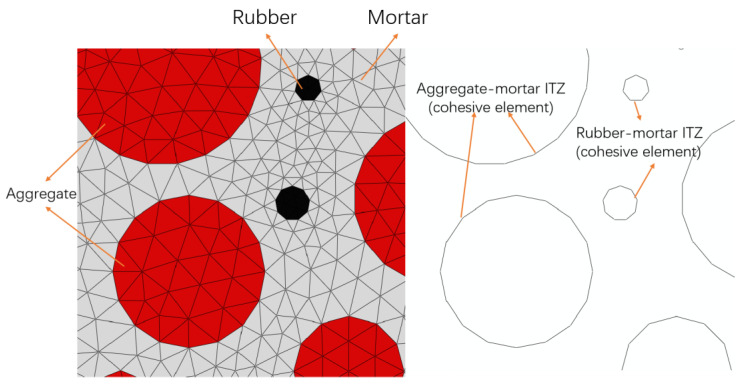
Zero-thickness cohesive element generated in the ITZ.

**Figure 3 polymers-16-01579-f003:**
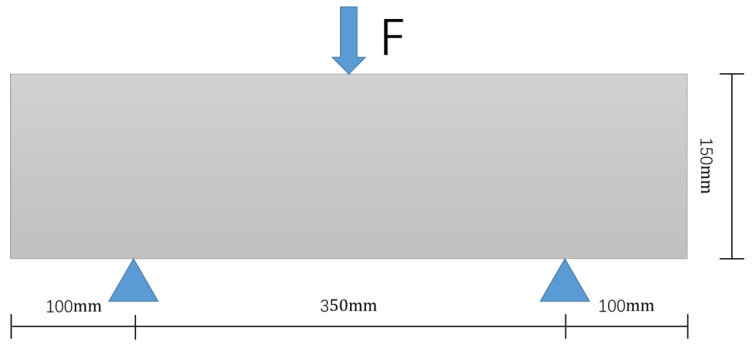
Load position and support position.

**Figure 4 polymers-16-01579-f004:**
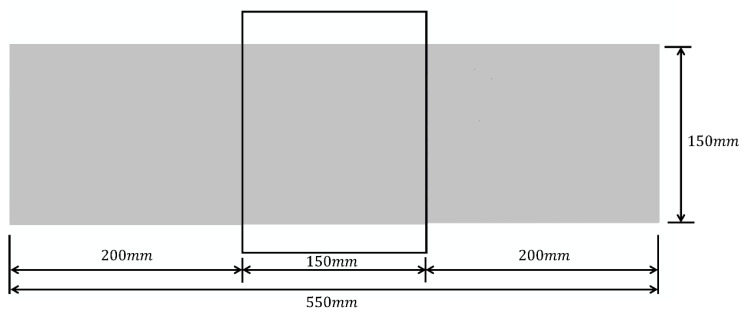
The location of SDEG interception.

**Figure 5 polymers-16-01579-f005:**
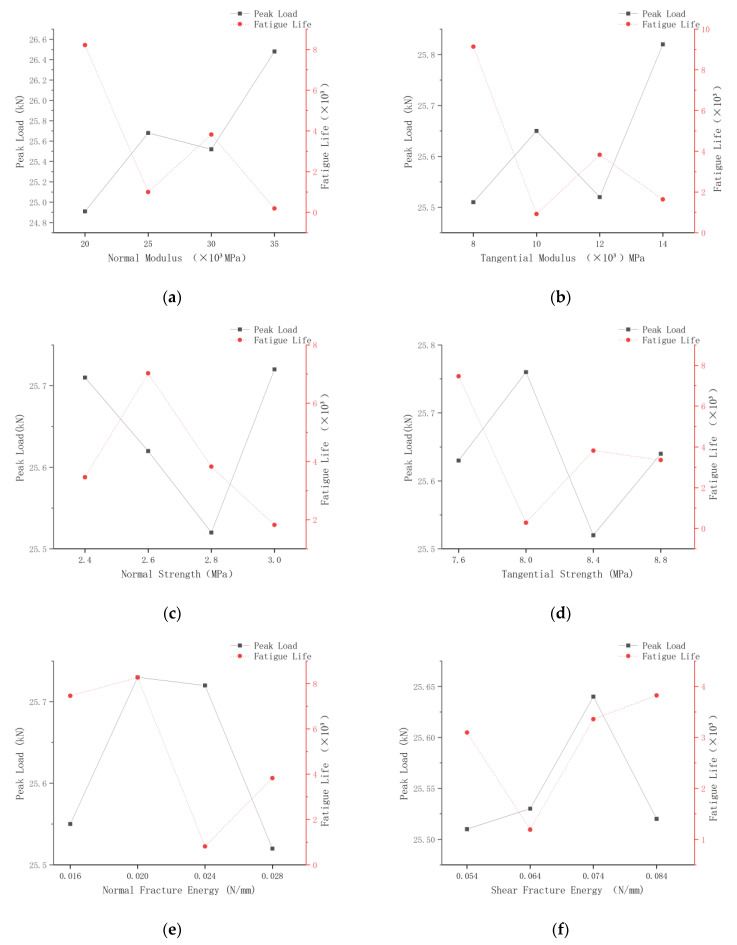
Peak load and fatigue life corresponding to six cohesive parameters: (**a**) normal modulus; (**b**) tangential modulus; (**c**) normal strength; (**d**) tangential strength; (**e**) normal fracture energy; (**f**) tangential fracture energy.

**Figure 6 polymers-16-01579-f006:**
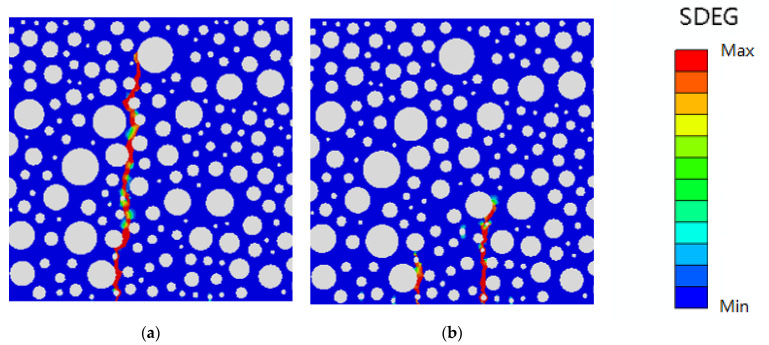
SDEG with different normal modulus for static pressure load: (**a**) 25,000 MPa; (**b**) 35,000 MPa.

**Figure 7 polymers-16-01579-f007:**
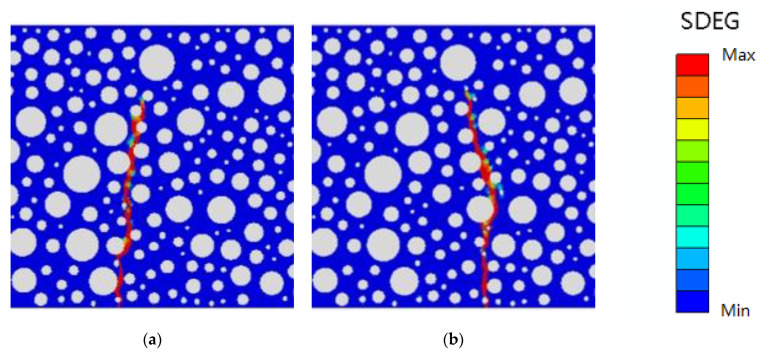
SDEG with different normal modulus for fatigue load: (**a**) 25,000 MPa; (**b**) 35,000 MPa.

**Figure 8 polymers-16-01579-f008:**
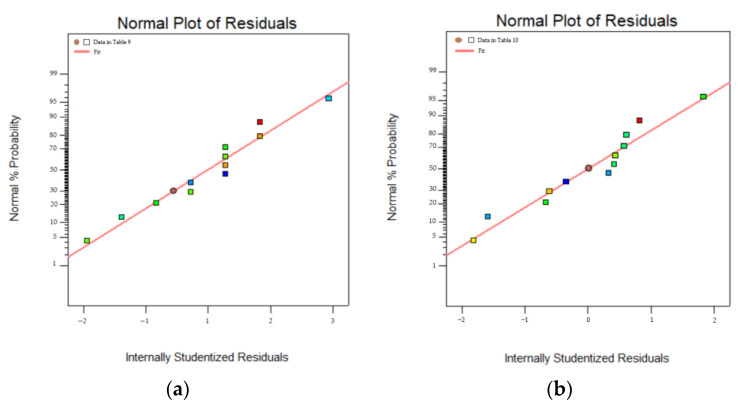
Residual normal distribution: (**a**) peak load; (**b**) fatigue life.

**Figure 9 polymers-16-01579-f009:**
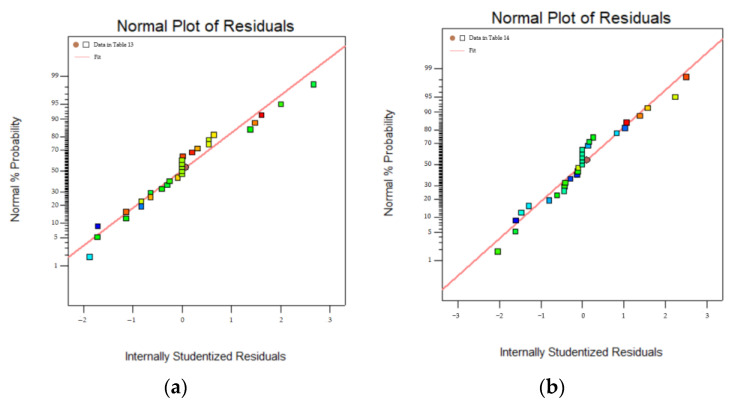
Residual normal distribution(quadratic correlation): (**a**) peak load; (**b**) fatigue life.

**Figure 10 polymers-16-01579-f010:**
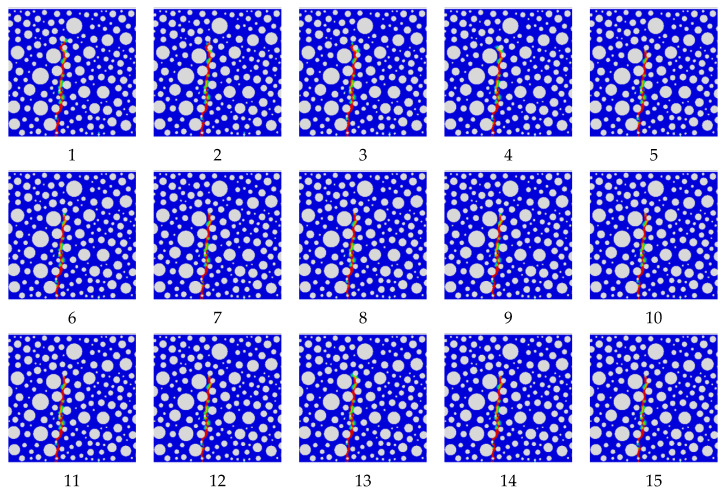
The 29 damage contour map corresponding to the design parameters in Table 12 for static pressure load.

**Figure 11 polymers-16-01579-f011:**
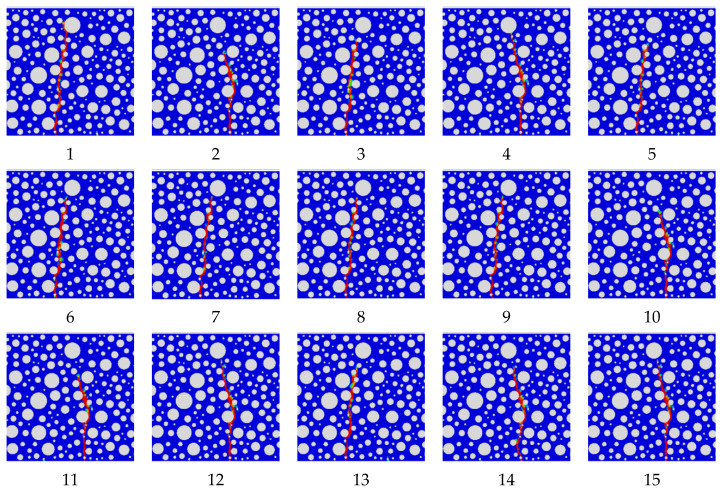
The 29 damage contour map corresponding to the design parameters in Table 12 for fatigue load.

**Figure 12 polymers-16-01579-f012:**
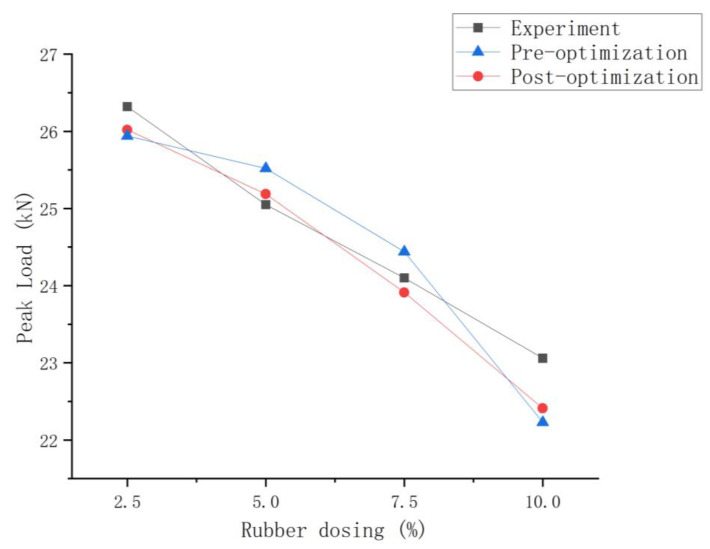
Peak Load with rubber dosing of 2.5%, 5%, 7.5%, and 10%.

**Figure 13 polymers-16-01579-f013:**
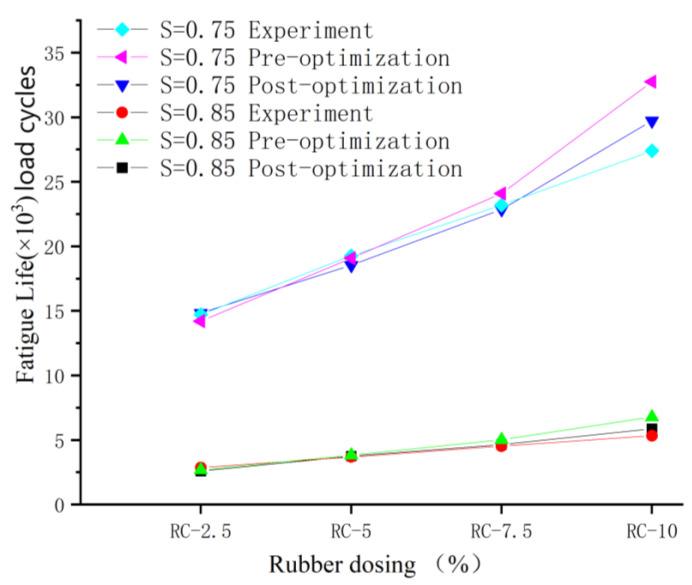
Fatigue life with rubber dosing of 2.5%, 5%, 7.5%, and 10%.

**Figure 14 polymers-16-01579-f014:**
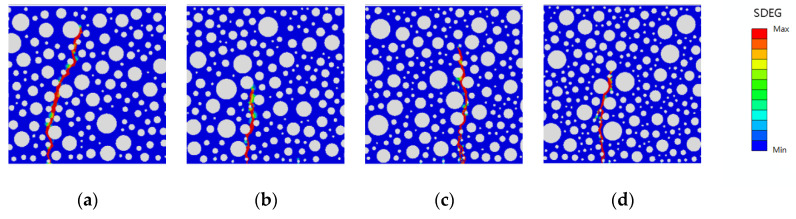
SDEG for static pressure load with rubber dosing: (**a**) 2.5%, (**b**) 5%, (**c**) 7.5%, and (**d**) 10%.

**Figure 15 polymers-16-01579-f015:**
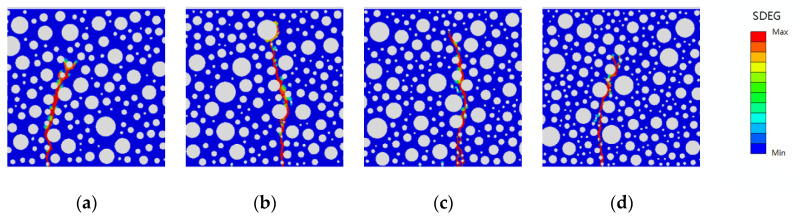
SDEG for fatigue load at S = 0.85 with rubber dosing: (**a**) 2.5%, (**b**) 5%, (**c**) 7.5%, and (**d**) 10%.

**Figure 16 polymers-16-01579-f016:**
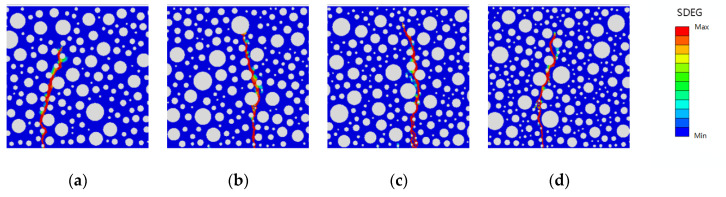
SDEG for fatigue load at S = 0.75 with rubber dosing: (**a**) 2.5%, (**b**) 5%, (**c**) 7.5%, and (**d**) 10%.

**Table 1 polymers-16-01579-t001:** Aggregate size distribution.

Rubber Replacement Rate (%)	5–10 mm (mm^2^)	10–15 mm (mm^2^)	15–20 mm (mm^2^)	Rubber (mm^2^)
5	12,547	8851	5612	1537

**Table 2 polymers-16-01579-t002:** Cohesive parameters of ITZ.

Type	Normal Strength (MPa)	Tangential Strength (MPa)	Normal Fracture Energy (N/mm)	Shear Fracture Energy (N/mm)
Aggregate-mortar ITZ	3.1	9	0.03	0.09
rubber-mortar ITZ	2.8	8.4	0.028	0.084

**Table 3 polymers-16-01579-t003:** Experimental and simulated peak load.

Experimental Peak Load (kN)	Simulated Peak Load (kN)	Error (%)
25.05	25.52	1.88

**Table 4 polymers-16-01579-t004:** Experimental and simulated fatigue life.

Type	S = 0.75	S = 0.85
Experiment (load cycles)	18,555	3671
Simulation (load cycles)	19,081	3824

**Table 5 polymers-16-01579-t005:** Values of cohesive parameters.

Normal Modulus (MPa)	Tangential Modulus (MPa)	Normal Strength (MPa)	Tangential Strength (MPa)	Normal Fracture Energy (N/mm)	Shear Fracture Energy (N/mm)
20,000	8000	2.4	7.6	0.016	0.054
25,000	10,000	2.6	8	0.02	0.064
30,000	12,000	2.8	8.4	0.024	0.074
35,000	14,000	3	8.8	0.028	0.084

**Table 6 polymers-16-01579-t006:** Standardized group of cohesive parameters.

Normal Modulus (MPa)	Tangential Modulus (MPa)	Normal Strength (MPa)	Tangential Strength (MPa)	Normal Fracture Energy (N/mm)	Shear Fracture Energy (N/mm)
30,000	12,000	2.8	8.4	0.024	0.074

**Table 7 polymers-16-01579-t007:** Value range of cohesive parameters.

Normal Modulus (MPa)	Tangential Modulus (MPa)	Normal Strength (MPa)	Tangential Strength (MPa)	Normal Fracture Energy (N/mm)	Shear Fracture Energy (N/mm)
27,000	11,000	2.5	7.8	0.022	0.07
32,000	13,000	2.9	8.6	0.03	0.09

**Table 8 polymers-16-01579-t008:** Results of Plackett–Burman design.

Normal Modulus (MPa)	Tangential Modulus (MPa)	Normal Strength (MPa)	Tangential Strength (MPa)	Normal Fracture Energy (N/mm)	Shear Fracture Energy (N/mm)	Peak Load (kN)	Fatigue Life (Load Cycles)
32,000	13,000	2.5	7.8	0.02	0.09	25.33	3071
27,000	11,000	2.5	7.8	0.02	0.07	25.02	7812
32,000	13,000	2.5	8.6	0.03	0.09	25.33	3087
32,000	11,000	2.9	8.6	0.02	0.09	25.12	6889
32,000	13,000	2.9	7.8	0.02	0.07	25.4	1744
27,000	13,000	2.9	7.8	0.03	0.09	25.23	4836
27,000	13,000	2.5	8.6	0.03	0.07	25.22	5037
32,000	11,000	2.9	8.6	0.03	0.07	25.19	5634
32,000	11,000	2.5	7.8	0.03	0.07	25.18	5882
27,000	11,000	2.5	8.6	0.02	0.09	24.97	9733
27,000	13,000	2.9	8.6	0.02	0.07	25.22	4913
27,000	11,000	2.9	7.8	0.03	0.09	25.09	8178

**Table 9 polymers-16-01579-t009:** ANOVA of peak load.

Variables	Sum of Squares of Variables	Variable Degree of Freedom	Mean Square	F	*p*
Normal Modulus	0.053	1	0.053	133.33	<0.0001
Tangential Modulus	0.11	1	0.11	280.33	<0.0001
Normal Strength	3.33 $\times \;10^{-3}$	1	3.33 $\times \;10^{-3}$	8.33	0.0343
Tangential Strength	3.33 $\times \;10^{-3}$	1	3.33 $\times \;10^{-3}$	8.33	0.0343
Normal Fracture Energy	2.7 $\times \;10^{-3}$	1	2.7 $\times \;10^{-3}$	6.75	0.0484
Shear Fracture Energy	2.13 $\times \;10^{-3}$	1	2.13 $\times \;10^{-3}$	5.33	0.069

**Table 10 polymers-16-01579-t010:** ANOVA of fatigue life.

Variables	Sum of Squares of Variables	Variable Degree of Freedom	Mean Square	F	*p*
Normal Modulus	1.68 $\times \;10^{7}$	1	1.68 $\times \;10^{7}$	197.29	<0.0001
Tangential Modulus	3.83 $\times \;10^{7}$	1	3.83 $\times \;10^{7}$	449.64	<0.0001
Normal Strength	4.91 $\times \;10^{5}$	1	4.91 $\times \;10^{5}$	5.77	0.0615
Tangential Strength	1.18 $\times \;10^{6}$	1	1.18 $\times \;10^{6}$	13.9	0.0136
Normal Fracture Energy	1.9 $\times \;10^{5}$	1	1.9 $\times \;10^{5}$	2.22	0.196
Shear Fracture Energy	1.9 $\times \;10^{6}$	1	1.9 $\times \;10^{6}$	22.27	0.0052

**Table 11 polymers-16-01579-t011:** Design parameters and levels for response surface method.

Normal Modulus (MPa)	Tangential Modulus (MPa)	Normal Strength (MPa)	Tangential Strength (MPa)
27,000	11,000	2.5	7.8
29,500	12,000	2.7	8.2
32,000	13,000	2.9	8.6

**Table 12 polymers-16-01579-t012:** Results of RSM.

Group	Normal Modulus (MPa)	Tangential Modulus (MPa)	Normal Strength (MPa)	Tangential Strength (MPa)	Peak Load (kN)	Fatigue Life (Load Cycles)
1	27,000	12,000	2.5	8.2	25.018	6591
2	27,000	12,000	2.7	8.6	25.223	3791
3	27,000	13,000	2.7	8.2	25.13	5057
4	32,000	13,000	2.7	8.2	25.215	4897
5	32,000	11,000	2.7	8.2	25.306	2654
6	29,500	11,000	2.7	8.6	25.243	3517
7	32,000	12,000	2.7	8.6	25.294	2814
8	29,500	12,000	2.9	7.8	25.257	3326
9	29,500	12,000	2.9	8.6	25.317	2499
10	29,500	12,000	2.7	8.2	25.2	4101
11	32,000	12,000	2.9	8.2	25.283	2968
12	29,500	12,000	2.7	8.2	25.2	4101
13	29,500	13,000	2.5	8.2	25.126	5115
14	27,000	11,000	2.7	8.2	25.132	5034
15	29,500	12,000	2.7	8.2	25.2	4101
16	32,000	12,000	2.5	8.2	25.147	5822
17	29,500	11,000	2.7	7.8	25.161	4632
18	29,500	13,000	2.7	8.6	25.145	5857
19	27,000	12,000	2.7	7.8	25.163	4608
20	29,500	12,000	2.5	7.8	24.974	7188
21	29,500	12,000	2.7	8.2	25.2	4101
22	29,500	11,000	2.5	8.2	25.146	4843
23	29,500	12,000	2.7	8.2	25.2	4101
24	29,500	13,000	2.7	7.8	25.045	6219
25	29,500	13,000	2.9	8.2	25.237	3599
26	27,000	12,000	2.9	8.2	25.218	6860
27	32,000	12,000	2.7	7.8	25.274	4088
28	29,500	11,000	2.9	8.2	25.257	3328
29	29,500	12,000	2.5	8.6	25.15	4784

**Table 13 polymers-16-01579-t013:** ANOVA of peak load(quadratic correlation).

Variables	Sum of Squares of Variables	Variable Degree of Freedom	Mean Square	F	*p*
Normal Modulus _(A)	0.034	1	0.034	17.68	0.0009
Tangential Modulus _(B)	9.46 $\times \;10^{-3}$	1	9.46 $\times \;10^{-3}$	4.98	0.0425
Normal Strength _(C)	0.085	1	0.085	44.55	<0.0001
Tangential Strength _(D)	0.02	1	0.02	10.44	0.006
AB	1.98 $\times \;10^{-3}$	1	1.98 $\times \;10^{-3}$	1.04	0.3247
AC	1.02 $\times \;10^{-3}$	1	1.02 $\times \;10^{-3}$	0.54	0.475
AD	4 $\times \;10^{-4}$	1	4 $\times \;10^{-4}$	0.21	0.6534
BC	0	1	0	0	1
BD	1.64 $\times \;10^{-5}$	1	1.64 $\times \;10^{-5}$	8.42 $\times \;10^{-3}$	0.9282
CD	3.36 $\times \;10^{-3}$	1	3.36 $\times \;10^{-3}$	1.77	0.2046
A^2^	1.29 $\times \;10^{-3}$	1	1.29 $\times \;10^{-3}$	0.68	0.4244
B^2^	1.91 $\times \;10^{-3}$	1	1.91 $\times \;10^{-3}$	1.01	0.3329
C^2^	2.61 $\times \;10^{-3}$	1	2.61 $\times \;10^{-3}$	1.37	0.2612
D^2^	1.2 $\times \;10^{-4}$	1	1.2 $\times \;10^{-4}$	0.063	0.8057

**Table 14 polymers-16-01579-t014:** ANOVA of fatigue life(quadratic correlation).

Variables	Sum of Squares of Variables	Variable Degree of Freedom	Mean Square	F	*p*
Normal Modulus _(A)	6.31 $\times \;10^{6}$	1	6.31 $\times \;10^{6}$	7.41	0.0165
Tangential Modulus _(B)	3.78 $\times \;10^{6}$	1	3.78 $\times \;10^{6}$	4.44	0.0535
Normal Strength _(C)	1.15 $\times \;10^{7}$	1	1.15 $\times \;10^{7}$	13.55	0.0025
Tangential Strength _(D)	3.85 $\times \;10^{6}$	1	3.85 $\times \;10^{6}$	4.53	0.0516
AB	1.23 $\times \;10^{6}$	1	1.23 $\times \;10^{6}$	1.45	0.2488
AC	2.44 $\times \;10^{6}$	1	2.44 $\times \;10^{6}$	2.87	0.1126
AD	52,212.25	1	52,212.25	0.061	0.8079
BC	0.25	1	0.25	2.94 $\times \;10^{-7}$	0.9996
BD	1.42 $\times \;10^{5}$	1	1.42 $\times \;10^{5}$	0.17	0.6893
CD	6.22 $\times \;10^{5}$	1	6.22 $\times \;10^{5}$	0.73	0.407
A^2^	4.40 $\times \;10^{5}$	1	4.40 $\times \;10^{5}$	0.52	0.484
B^2^	2.76 $\times \;10^{5}$	1	2.76 $\times \;10^{5}$	0.32	0.5778
C^2^	1.48 $\times \;10^{6}$	1	1.48 $\times \;10^{6}$	1.74	0.2083
D^2^	4995	1	4995	5.87 $\times \;10^{-3}$	0.94

**Table 15 polymers-16-01579-t015:** Optimized cohesive parameters and expected targets.

Normal Modulus (MPa)	Tangential Modulus (MPa)	Normal Strength (MPa)	Tangential Strength (MPa)	Expected Peak Load (kN)	Expected Fatigue Life (Load Cycles)
29,300	11,000	2.57	8.6	25.194	3684

**Table 16 polymers-16-01579-t016:** Comparative results.

	Peak Load (kN)	Fatigue Life (Load Cycles)
Expected value	25.194	3684
Experimental value	25.05	3671
Simulated value	25.187	3744

**Table 17 polymers-16-01579-t017:** Comparison of peak loads with different rubber dosing.

	RC–2.5	RC–5	RC–7.5	RC–10
Experiment (kN)	26.32	25.05	24.1	23.06
Pre-optimization (kN)	25.94	25.52	24.44	22.23
Pre-optimization error	−1.44%	1.88%	1.41%	−3.6%
Post-optimization (kN)	26.02	25.187	23.913	22.41
Post-optimization error	−1.14%	0.55%	−0.78%	−2.82%

**Table 18 polymers-16-01579-t018:** Comparison of fatigue life with different rubber dosing at the stress level of 0.85.

	RC–2.5	RC–5	RC–7.5	RC–10
Experiment (load cycles)	2870	3684	4521	5330
Pre-optimization (load cycles)	2678	3824	5018	6779
Pre-optimization error	−6.69%	3.8%	10.99%	27.2%
Post-optimization (load cycles)	2588	3744	4651	5861
Post-optimization error	−9.83%	1.98%	2.88%	9.97%

**Table 19 polymers-16-01579-t019:** Comparison of fatigue life with different rubber dosing at the stress level of 0.75.

	RC–2.5	RC–5	RC–7.5	RC–10
Experiment (load cycles)	14,706	19,265	23,188.5	27,403
Pre-optimization (load cycles)	14,208	19,081	24,085	32,742
Pre-optimization error	−3.39%	−0.96%	3.87%	19.48%
Post-optimization (load cycles)	14,823	18,534	22,859	29,724
Post-optimization error	0.80%	−3.79%	−1.42%	8.47%

## Data Availability

Data are contained within the article.

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
