# Peer review of "Optimization of Cohesive Parameters in the Interfacial Transition Zone of Rubberized Concrete Based on the Response Surface Method"

_polymers, 2024, doi:10.3390/polym16111579_

Round 1

Reviewer 1 Report

Comments and Suggestions for Authors

The manuscript "Optimization of Cohesive Parameters in the Interfacial Transition Zone of Rubberized Concrete Based on Response Surface Method Special Issue: Preparation and Application of Rubber Composites" contains many inaccuracies that require clarification and correction. The authors should write what they considered when selecting the six cohesion parameters. The main problem is the statement that the normal strength is lower than the tangential strength even though the normal modulus is larger than the tangential modulus. This requires some explanation.

General remarks

1. In the article, the authors should specify the scope of the micro-scale, fine-scale and macro-scale they describe.

2. It should be explained why the authors use two terms: meso-scale and fine-scale; or standardize.

3. The authors should write in the manuscript why they chose a 1-point load and not, for example, a 3-point load and why they chose the span to beam height ratio of 350/150 mm.

4. 179 line: The manuscript should explain why the load levels S were adopted?

5. It should be explained what the dimensions of the samples were and what was the design in the experimental tests in Table 3

6. It should be explained why Tangential strength is greater than the normal strength, even though the normal modulus is greater than the tangential modulus (Table 15).

Specific remarks

7. 120 line: write what exactly D means.

8. 124 line: Please write what is the third dimension (0) of the adopted model.

9. 129 line: Authors should explain what "Phyton" is.

10. 146 line: Write "with reduced compressive (tensile) strength…

11. 149 line: Please write what type of strength.

12. 158 line: Please write what type of the normal strength.

13. 168, 171 lines: You should write supports, not "constrains".

14. 192 line: Please write in what units the quantities are presented in Tab.4.

15. 212 line: Please write the type of normal strength.

16. 216 line: It should be explained why this set of six parameters was selected. Tab.5.

17. 232 line: Write "kN" and 1485 load cycles.

18. 242 line: Explain the abbreviation SDEG.

19. 272 line, Tab.8: What does the notation Tangential /..0Modulus mean? . It should be “Peak Load kN” .

20. 274- 277 line: Please write "kN". There should be, for example, 1744 load cycles.

21. 328 line: The abbreviation RSM should be explained.

22. 336 line: Please write kN, not KN.

23. 404 line: Should be "kN" not "KN"

24. 427 line, Tab.17: It should be "kN" on the ordinate axis, not "KN".

25. 429 line, Tab.17: It should be on the ordinate axis "kN" and not "KN".

26. 540 line: Conclusions should be bulleted.

27. 569 line: Please explain what [J] and [D] mean in references. You should format your reference list properly.

I recommend an in-depth review of the manuscript, including comments, to make it an article suitable for publication in the Polymers.

In its current state, the article should not be published.

Comments on the Quality of English Language

Minor editing of English language required.

Reviewer 2 Report

Comments and Suggestions for Authors

1. Did the authors use “cohesive surface”, instead of the “cohesive element”, in Abaqus to simulate the zero-thickness interfacial transition zone (ITZ)? Did the authors set the cohesive behaviour in all ITZs, including aggregate-mortar ITzs and rubber-mortar ITZs, in the FE models? The wording should be more precise.

2. How did the authors simulate the mortar, aggregates, and rubbers? Are they all elastic in the FE models? Please describe the numerical models in detail.

3. It is surprising that the stop criterion of the analysis is the divergence during the calculation process. Did the authors use “static, general” with nonlinear geometric analysis in the analyzing step? Have the authors tried other analysis steps, such as "dynamic, implicit", or modified the point load by a distributed load with a small bearing plate? The numerical results are doubtful without showing the load-displacement curve of the three-point bending test.

4. What are the failure mechanisms of the specimen in the actual three-point bending tests? How did the authors demonstrate the accuracy of their numerical simulations? The peak load of the three-point bending test alone is not enough to prove accuracy.

5.  How did the authors simulate the fatigue damage in the FE models? What are the failure criteria in the FE models?

6. The authors should describe what the response surface function is in this study.

7. In Figures 10 and 11, there is only one crack in each specimen. Did any experimental images support these results? In my opinion, it is not reasonable.

Comments on the Quality of English Language

None

Round 2

Reviewer 1 Report

Comments and Suggestions for Authors

Compared to the previous version of the article, they have introduced corrections that partially take into account the reviewer's suggestions. However, there are still serious inaccuracies that require necessary correction.

In table 11 Normal modulus is 27000 MPa. Why in table 12 Normal modulus 2.7 MPa. Moreover, this applies to all Normal modulus and Tangential modulus values in the table. 12.

Ad. 3-4 The authors still have not answered the questions regarding the selection of the load, cross-section and load levels S. However, they refer to the literature [39]. You should write why these particular quantities are appropriate for your research.

Ad. 9 The phrase Python appears on line 130 (135) of the article.

Ad 27 It should be "Journal".

Comments on the Quality of English Language

Minor editing of English language required.
